# A comprehensive update to the *Mycobacterium tuberculosis* H37Rv reference genome

Poonam Chitale [1,2], Alexander D. Lemenze[3], Emily C. Fogarty [4,5], Avi Shah[1,6], Courtney Grady[1,2], Aubrey R. Odom-Mabey [7,8], W. Evan Johnson[1,2,9], Jason H. Yang [1,6], A. Murat Eren[10,11], Roland Brosch [12], Pradeep Kumar[1,2] & David Alland [1,2] ✉

H37Rv is the most widely used *Mycobacterium tuberculosis* strain, and its genome is globally used as the *M. tuberculosis* reference sequence. Here, we present Bact-Builder, a pipeline that uses consensus building to generate complete and accurate bacterial genome sequences and apply it to three independently cultured and sequenced H37Rv aliquots of a single laboratory stock. Two of the 4,417,942 base-pair long H37Rv assemblies are 100% identical, with the third differing by a single nucleotide. Compared to the existing H37Rv reference, the new sequence contains ~6.4 kb additional base pairs, encoding ten new regions that include insertions in PE/PPE genes and new paralogs of *esxN* and *esxJ*, which are differentially expressed compared to the reference genes. New sequencing and de novo assemblies with Bact-Builder confirm that all 10 regions, plus small additional polymorphisms, are also present in the commonly used H37Rv strains NR123, TMC102, and H37Rv1998. Thus, Bact-Builder shows promise as an improved method to perform accurate and reproducible de novo assemblies of bacterial genomes, and our work provides important updates to the primary *M. tuberculosis* reference genome.

*Mycobacterium tuberculosis* is estimated to infect roughly a quarter of the world's population and was the second leading cause of death from an infectious disease after SARS-CoV-2 in 2020[1]. In 1998, Cole et al. sequenced the first complete *M. tuberculosis* genome using *M. tuberculosis* H37Rv, a strain that was first isolated from a patient with pulmonary tuberculosis in 1905[2–4]. Although the original H37Rv strain has been lost, H37Rv strains TMC 102 (ATCC 27294) and NR-123 (ATCC 25618), both isolated from the same patient in separate years are frequently used in H37Rv studies[5]. H37Rv remains the most widely used *M. tuberculosis* strain for laboratory experimentation, and the 1998

[1]Ray V. Lourenco Center for the Study of Emerging and Re-emerging Pathogens, Rutgers University – New Jersey Medical School, Newark, NJ, USA. [2]Public Health Research Institute, Rutgers University – New Jersey Medical School, Newark, NJ, USA. [3]Department of Pathology, Immunology and Laboratory Medicine, New Jersey Medical School, Rutgers—The State University of New Jersey, Newark, NJ, USA. [4]Department of Medicine, University of Chicago, Chicago, IL, USA. [5]Committee on Microbiology, University of Chicago, Chicago, IL, USA. [6]Department of Microbiology, Biochemistry and Molecular Genetics, Rutgers University- New Jersey Medical School, Newark, NJ, USA. [7]Division of Computational Biomedicine, Boston University School of Medicine and Bioinformatics Program, Boston University, Boston, MA, USA. [8]Bioinformatics Program, Boston University, Boston, MA, USA. [9]Center for Data Science, Rutgers University – New Jersey Medical School, Newark, NJ, USA. [10]Helmholtz Institute for Functional Marine Biodiversity (HIFMB), Oldenburg, Germany. [11]Bay Paul Center, Marine Biological Laboratory, Woods Hole, MA, USA. [12]Institut Pasteur, Université Paris Cité, Unit for Integrated Mycobacterial Pathogenomics, CNRS UMR 6047 Paris, France. ✉e-mail: allandda@njms.rutgers.edu

*H37Rv whole genome sequence is widely accepted as the M. tuberculosis reference sequence.*

Bacterial whole genome sequencing (WGS) is now commonly applied to a variety of scientific disciplines. WGS is usually performed with next-generation sequencing (NGS) tools that generate short (75–300 bp) sequencing reads. These reads can then be rapidly de novo assembled to create draft genomes. However, these assemblies are typically fragmented, harbor mapping artifacts, and contain mis-annotated gene calls[6]. This problem is largely addressed by mapping reads to a known and extensively curated reference genome such as H37Rv, generating a reference-based assembly. Despite its widespread use, uncertainties remain regarding the completeness of the currently available H37Rv sequence. Repetitive sequences, gene duplications and gene inversions are known confounders of whole genome assembly tools[7], and *M. tuberculosis* is particularly subject to these limitations given that approximately 10% of the *M. tuberculosis* genome is comprised of highly repetitive genes[8]. Known differences among different H37Rv isolates further confounds this issue[5]. These findings suggest that a reanalysis of the *M. tuberculosis* H37Rv reference using more accurate genome assembly tools might reveal important new elements of this genome that could impact all studies that use H37Rv as the genome reference.

The development of third-generation sequencing tools such as single molecule sequencing technology that produces (1 kb ->100 kb) long reads has helped to successfully resolve repetitive or gapped gene regions, enabling the generation of complete and closed microbial genomes assemblies[7,9,10]. These new technologies can substantially improve the accuracy of reference genomes and can enable direct comparisons among different bacterial genomes without first performing a reference-based assembly. Direct genome comparisons can be particularly useful when organisms contain "accessory" sequences that are not present in a standard reference genome. By definition, accessory sequences cannot be mapped to a reference, and this results in their exclusion from any reference-based analysis. Several tools have been designed to de novo assemble NGS and third-generation sequencing data[11–17]; however, the variety of these tools and algorithms leads to inherent differences in their outputs[16–19], and there is no standard approach for de novo bacterial genome assembly. Existing programs are largely limited by a long- or short-read-only approach, and/or the use of a specific single program to assemble the genome[20–22]. Hybrid assembly tools bridge this gap but typically use short-read sequencing to build an initial scaffold followed by long reads on top to create a draft genome[16,17]. These approaches often ignore the shortcomings of individual assemblers and those that rely on a long read-only approach must contend with the large number of SNPs and small indels inherently present in long-read sequencing and not caught by older base calling tools. Furthermore, very few programs or pipelines exist that enable end-to-end generation of highly accurate gap-closed genomes starting with raw sequencing data.

In our efforts to readdress the *M. tuberculosis* H37Rv sequence and support direct comparisons among de novo assembled genomes of clinical *M. tuberculosis* strains, we developed a new pipeline that allows us to generate highly accurate complete whole genome sequences using a de novo assembly approach. Named Bact-Builder, this tool uses a long-read consensus-based assembly approach, which is then followed by long and short-read polishing.

Here, we first demonstrate the virtual 100% accuracy and reproducibility of this approach when sequencing three separate H37Rv cultures, as well as the ability of the assembly tool to reveal ~6.4 kb of new sequence that is absent from the GenBank (NC_000962.3) published reference, including 10 major regions of difference. Together, this work reveals an important update to the H37Rv references sequence and shows the utility of Bact-Builder for assembling highly accurate bacterial genomes, both for use as genomic references or applied to other metagenomic and pangenome studies of bacterial species.

## Results

### Building and validating Bact-Builder

Bact-Builder (Fig. 1a) was designed to generate highly accurate gap-closed de novo bacterial genomes. Bact-Builder steps were first designed using in silico-generated reads created with BadReads[23] and ART[24], which removed common experimental variables that can affect sequencing output and quality (Supplementary Note 1, Figure S2a–d). Further testing was then performed by experimentally sequencing three independent cultures of the same H37Rv stock, extracting genomic DNA, preparing sequencing libraries and sequencing each culture separately. Sequencing was performed on both Oxford Nanopore Technologies (ONT) MinION and Illumina NovaSeq (paired-end 2×150) platforms (Table S1), and the raw sequencing data was used to assemble each of the three replicates (H37Rv.1-3) using Bact-Builder (Fig. 1a, Figure S2 e-h). We first tested the accuracy and reproducibility of four commonly used long-read assembly tools, Canu, Flye, Miniasm and Raven run in triplicate for all samples. We found significant differences between assemblers across all three sequenced H37Rv samples (Fig. 1b–d; Figure S3 a-b). Unlike the analysis with our in silico data, the analysis of the genomes reconstructed from the three in vitro sequenced samples using a pan-genomics approach revealed critical differences between the reporting of individual assemblers both within a sample and across multiple samples (Fig. 1e; Figure S3 c-d, Table S3). These differences included missing core genes and spurious accessory genes compared to the reference genome, suggesting that the existing tools to process long-read sequencing data are not suitable to reconstruct microbial genomes when accuracy and reproducibility are critical concerns. We further tested the impact of polishing the individual assemblies and saw that although polishing improved some assembly sizes, there remained inconsistencies across assembly sizes producing genomes of substantially different lengths and incorrectly called indels (Supplementary Note 2, Table S4, Table S5, Figure S5).

We worked to reconcile the differences observed across assemblers using Trycycler[19] a program that generates a consensus sequence from the outputs of the four individual assemblers. Trycycler produced contigs that were more similar to the reference than any individual assembly and further polishing with both long and short reads generated even more accurate assemblies (Fig. 1c), although some remained too fragmented or could not be reconciled and were dropped by the program (Fig. 1b). Examining the contribution of sequence coverage depth to the quality of the final Bact-Builder result, we compared Bact-Builder outputs to the H37Rv.1-3 samples using data subsetted to 30x, 50x 100x, 250x, and 500x average read depth. Unlike in silico generated reads (Figure S1b), we found that 30x coverage did not generate a consensus assembly for 2 replicates (H37Rv.2, H37Rv.3) because 3 out of 4 assemblers could not be reconciled and had to be dropped (Fig. 1b). At 50x coverage and above, we did not observe significant variations in genome size (Fig. 1b). Using the >50x coverage, Dnadiff indicated that Bact-Builder's assembly of the three independent H37Rv samples produced were identical in size and whole genome sequence (No SNPs were found between H37Rv.1, H37Rv.2 and H37Rv.3) except for one sample (H37Rv.1) that was one nucleotide shorter than the other two (Table 1). Further investigation into the H37Rv.1 indel (compared to H37Rv.2 and H37Rv.3), showed that it was an assembly error (Supplementary Note 2, Figure S5). Updating relevant components of Bact-Builder corrected the single basepair indel (Supplementary Note 3, Figure S6). The complete updated genome of H37Rv.1 is reported here as H37Rv (new) and the full genome sequence has been deposited (Supplementary Data 2, 3; Genbank accession: CP110619).

## Corrections to the H37Rv genome

A comparison between H37Rv(new) and the H37Rv1998 reference strain genome (NC_000962.3) revealed ten regions (R) of difference (R1 – R10) found only in H37Rv(new) (Fig. 2). R1 was a 207 bp in-frame insertion in *PE_PGRS27* (*rv1450c*). R2 was a 1356 bp sequence that contained duplications of *rv3475* and *rv3474*, which together make up an IS6110 transposase. There are 7 copies of this region found in the H37Rv1998 and 8 found in H37Rv.1-3. In R3 a single complete copy of *PPE38* was replaced by an 2064 bp sequence that contained an upstream copy of *PPE38* and a downstream second truncated copy of *PPE38* (which we have named *PPE38a* or *rv2351c.2*). Between these two *PPE38* genes we found a new paralog of *esxN* (*rv1793*) which we have named *esxN.2* (*rv2351c.3*), and a new paralog of the *esxJ* (*rv1038c) and esxM* genes (*rv1792*) (*esxM is also called esxJ in M. tuberculosis CDC1551*)[8,25], which we have named *esxJ.3* (*rv2351c.4*). The R4, R5 and R6 sequences each contained increased numbers of tandem

duplications in intergenic regions. R7 was a 1728 bp in-frame insertion at the 3' end of *PPE54* (*rv3343c*), substantially increasing its size. R8 was a 9 bp insertion of a tandem duplication in *PE_PGRS51* (*rv3367*). R9 was a 579 bp in-frame insertion in the middle of *PE_PGRS54* (*rv3508*), which also significantly increased the size of the gene. R10 was a 111 bp in-frame insertion in the middle of *PE_PGRS57* (*rv3514*). In addition to the 10 regions, 109 SNPs and 35 indels were found between H37Rv(new) and the H37Rv1998 reference (Supplementary Data 1).

We evaluated the potential functional impact of amino acid differences between *esxN* and *esxN.2* or between *esxJ*, (*CDC551-esxJ/esxM*) and *esxJ.3* using PROVEAN[26], which suggested that the paralogs maintained an overall conservation of protein structure (Fig. 3a). PCR and subsequent Sanger sequencing of R3 revealed that the observed differences were real and were likely missed using the original shot-gun sequencing approach (Fig. 3b). We also PCR amplified H37Rv strains NR123, TMC102, TMC301 and H37Rv1998 and detected R3 in all three

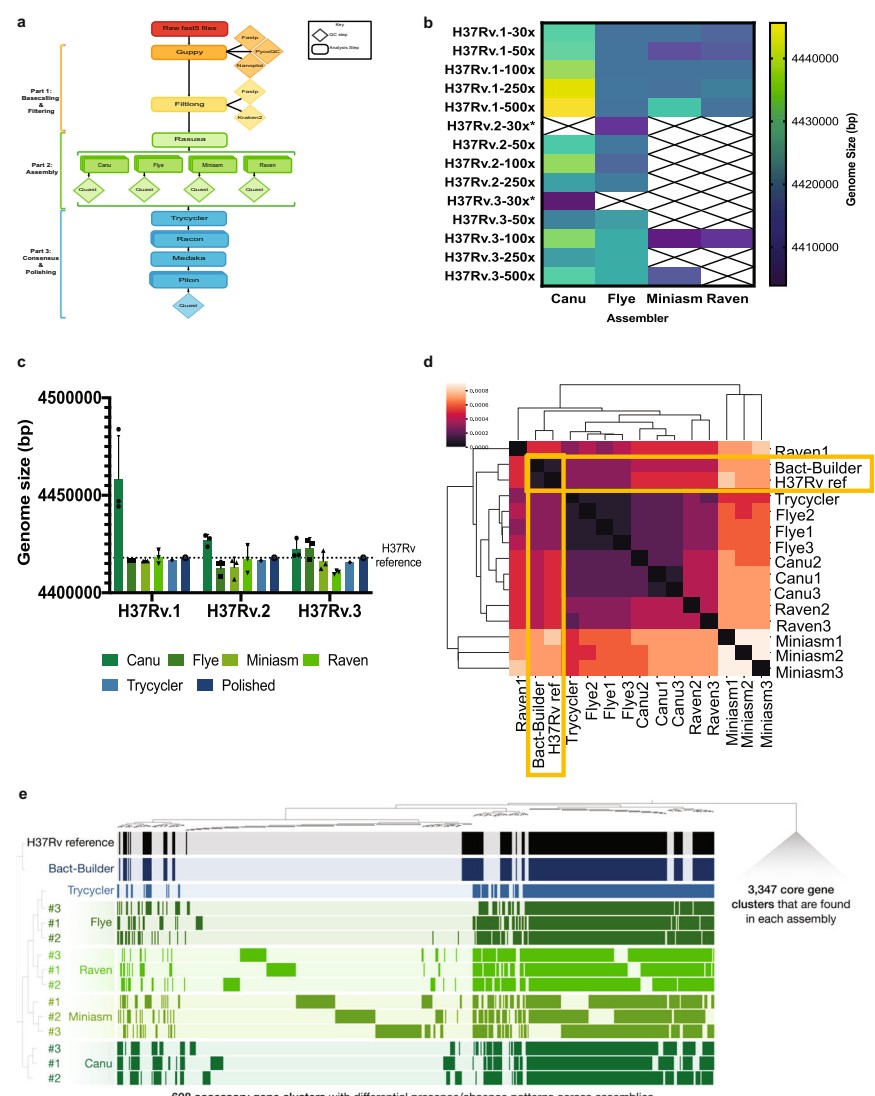

**Fig. 1 | Developing Bact-Builder. a** Pipeline overview. Bact-Builder takes raw fast5 sequencing data, files, assembles, generates a consensus, and polishes bacterial genomes. **b** Heatmap comparison of genome sizes of four de novo long read assemblers from laboratory stocks of H37Rv sequenced in triplicate (H37Rv.1-3). The sequence coverage sampled for each analysis is shown in each row on the Y axis. Boxes marked by an X indicate that the assemblies did not pass the Trycycler stage because they could not be reconciled with the other assemblies. * Indicates that 3 out of 4 assemblers could not be reconciled, necessitating that Trycycler was run with only 1 assembler. **c** Three replicates of laboratory stocks of H37Rv

(H37Rv.1-3), showed variability in size depending on assembler used, and consistent sizes when Trycycler was followed by polishing (Bact-Builder output). Dotted line indicates the size of the established H37Rv reference. Data are plotted as means ± SD. **d** Heatmap of hierarchical clustering of the distance using Euclidean average linkage clustering of differences between all assemblies for H37Rv.1, the Bact-Builder output and the published reference (H37Rv ref) determined by DNAdiff. **e** Anvi'o pangenome comparing gene clusters in the reference (H37Rv ref) and H37Rv.1 individual assemblies, Trycycler output and the Bact-Builder output.

**Table 1 | H37Rv.1-3 individual assembly results, and DNAdiff analysis comparing individual assemblies to the H37Rv reference (H37Rv ref)**

| Name | # of contigs | Size (bp) | SNP count[a] | Indel count[b] | RD count[c] |
|---|---|---|---|---|---|
| H37Rv ref | 1 | 4411532 | – | – | – |
| H37Rv.1 canu1 | 1 | 4446966 | 109 | 2005 | 10 |
| H37Rv.1 canu 2 | 3 | 4483900 | 110 | 1989 | 13 |
| H37Rv.1 canu 3 | 1 | 4444289 | 110 | 2019 | 10 |
| H37Rv.1 flye 1 | 1 | 4416736 | 123 | 1251 | 9 |
| H37Rv.1 flye 2 | 1 | 4416759 | 133 | 1234 | 9 |
| H37Rv.1 flye 3 | 1 | 4416479 | 126 | 1248 | 9 |
| H37Rv.1 miniasm 1 | 1 | 4416466 | 241 | 2961 | 9 |
| H37Rv.1 miniasm 2 | 1 | 4416251 | 240 | 2841 | 9 |
| H37Rv.1 miniasm 3 | 1 | 4416359 | 240 | 2841 | 9 |
| H37Rv.1 raven 1 | 1 | 4414975 | 159 | 1909 | 11 |
| H37Rv.1 raven 2 | 1 | 4422017 | 140 | 1916 | 10 |
| H37Rv.1 raven 3 | 1 | 4418092 | 149 | 1835 | 11 |
| H37Rv.1 Trycycler | 1 | 4416834 | 110 | 1157 | 8 |
| H37Rv.1 Bact-Builder | 1 | 4417941 | 109 | 36 | 10 |
| H37Rv.2 canu1 | 1 | 4429379 | 113 | 2301 | 9 |
| H37Rv.2 canu 2 | 1 | 4427845 | 111 | 2323 | 9 |
| H37Rv.2 canu 3 | 1 | 4423645 | 110 | 2326 | 9 |
| H37Rv.2 flye 1 | 1 | 4408125 | 125 | 1386 | 10 |
| H37Rv.2 flye 2 | 1 | 4414937 | 133 | 1377 | 9 |
| H37Rv.2 flye 3 | 1 | 4414949 | 137 | 1377 | 9 |
| H37Rv.2 miniasm 1 | 5 | 4406947 | 137 | 1896 | 11 |
| H37Rv.2 miniasm 2 | 4 | 4415501 | 130 | 1246 | 12 |
| H37Rv.2 miniasm 3 | 4 | 4416498 | 125 | 1852 | 10 |
| H37Rv.2 raven 1 | 2 | 4424323 | 107 | 1727 | 24 |
| H37Rv.2 raven 2 | 1 | 4410082 | 197 | 1638 | 30 |
| H37Rv.2 raven 3 | 1 | 4410082 | 197 | 1638 | 30 |
| H37Rv.2 Trycycler | 1 | 4416448 | 114 | 1538 | 8 |
| H37Rv.2 Bact-Builder | 1 | 4417942 | 109 | 35 | 10 |
| H37Rv.3 canu 1 | 1 | 4428104 | 123 | 5436 | 10 |
| H37Rv.3 canu 2 | 1 | 4419097 | 121 | 5537 | 10 |
| H37Rv.3 canu 3 | 1 | 4419406 | 131 | 5465 | 9 |
| H37Rv.3 flye 1 | 4 | 4416882 | 201 | 2825 | 11 |
| H37Rv.3 flye 2 | 4 | 4425435 | 190 | 2846 | 15 |
| H37Rv.3 flye 3 | 4 | 4426660 | 196 | 2885 | 16 |
| H37Rv.3 miniasm 1 | 1 | 4421693 | 158 | 3350 | 11 |
| H37Rv.3 miniasm 2 | 1 | 4414787 | 173 | 3619 | 10 |
| H37Rv.3 miniasm 3 | 1 | 4412379 | 190 | 3609 | 12 |
| H37Rv.3 raven 1 | 1 | 4410598 | 164 | 3385 | 25 |
| H37Rv.3 raven 2 | 1 | 4411303 | 141 | 3134 | 21 |
| H37Rv.3 raven 3 | 1 | 4409434 | 188 | 3119 | 21 |
| H37Rv.3 Trycycler | 1 | 4415585 | 114 | 2400 | 8 |
| H37Rv.3 Bact-Builder | 1 | 4417942 | 109 | 37 | 10 |

[a]SNP count: Single Nucleotide Polymorphism count relative to H37Rv ref.
[b]Indel count: single base insertions or deletions count relative to H37Rv ref; RD count.
[c]Regions of Difference relative to the H37Rv reference.

strains (Fig. 3b). These results are also supported by previous studies which identified the genes found in our H37Rv R3 region in another strain of *M. tuberculosis and Mycobacterium marinum* and *Mycobacterium microti*, although using a different nomenclature[25,27–29].

Finally, we completely sequenced and used Bact-Builder to de novo assemble NR123, TMC102, and H37Rv1998. We again detected all 10 regions of difference confirming that R1–10 are true regions shared by all publicly available H37Rv isolates. A phylogenetic analysis of these strains demonstrated that H37Rv.1–3 were most closely related to TMC102, differing by only 4 SNPs (Fig. 4). We found that H37Rv1998 was most closely related to NR123, but NR123 still differed from H37Rv1998 by in-frame insertions in *rv2680* (conserved hypothetical protein), *rv3367* (*PE_PGRS51*) and intergenic regions, and by in-frame deletions in *rv0297* (*PE_PGRS5*), *rv3514* (*PE_PGRS57*) and in intergenic regions (Table 2, Fig. 4). Compared to NR123, TMC102 and H37Rv.1-.3 had two additional IS6110 transposon elements, and in-frame insertions in *rv0297* (*PE_PGRS5*), *rv2090* (Probable 5′−3′ exonuclease), *rv3367* (*PE_PGRS51*) and in intergenic regions (Table 2).

**All 10 H37Rv regions are expressed**
As further confirmation that none of the regions found in H37Rv(new) were caused by sequencing or assembly artifacts, we tested whether published H37Rv RNA sequencing studies contained reads that mapped to each region. This investigation also allowed us to determine whether the newly discovered *esxN.2* and *esxJ.3* genes were expressed differently compared to their *esxN* and *esxJ* paralogs as one possible indication of significant functional differences between these genes. We downloaded all public *M. tuberculosis* H37Rv RNA sequencing datasets on the NCBI SRA database. Of the 908 datasets available at the time of our analysis, 905 datasets used Illumina sequencing chemistry, passed our quality control metrics, and were therefore included in these analyses. We aligned these raw reads to both the existing H37Rv1998 reference genome and to H37Rv(new) and computed the feature counts for each genomic region of difference following batch correction. We discovered that raw RNA reads from public data sets did indeed map onto the newly discovered regions in H37Rv(new) (Figure S4). However, both *esxN.2* and *esxJ.3* showed significantly different levels of expression across the 905 datasets compared to their *esxN* and *esxJ* paralogs (Fig. 3c, d).

We also found that on average, 99.8% of the reads mapping to *esxN.2* and 46% of the reads mapping to *esxJ.3*, did not map back to H37Rv1998 (Figure S4k), indicating that *esxN.2* reads were likely discarded in previous studies and that *esxJ.3* expression was likely ascribed mistakenly to *esxJ.1*. Expression levels of both *esxN* paralogs and of both *esxJ* paralogs were only moderately correlated (r = 0.557; $p = 7.02e-75$ and r = 0.516; $p = 1.41e-65$ respectively), suggesting that the paralogs were under different regulatory control (Fig. 3e, f). Expression between *esxN.2* and *esxJ.3*, however, were strongly correlated (r = 0.788; $p = 2.04e-195$), likely because the genes are adjacent to each other (Fig. 4g).

## Discussion
The development of new NGS tools and pipelines to generate accurate complete genomes is vital to our understanding of drug resistance evolution, pathogenesis and virulence of clinically relevant pathogens such as *M. tuberculosis*. The first complete whole genome sequence of the H37Rv strain generated using shotgun sequencing was published in 1998[4] and ushered in a revolution in *M. tuberculosis* research. The published sequence led to the discovery of novel protein families responsible for fatty acid and polyketide biosynthesis, drug efflux pumps, PE/PPE proteins and transposon elements[4,5,30]. Minor divergences among subcultures of these strains leading to differences in SNPs, indels and transposable elements have also been described[5].

H37Rv is the workhorse of *M. tuberculosis* research and its Genbank sequence is widely used for aligning and analyzing DNA, RNA, and Tn-seq studies. Thus, the importance of establishing a fully accurate reference genome for this strain cannot be overstated. Our current study indicates that there are substantial regions of difference between the Genbank H37Rv reference sequence and what is likely to be the true complete genome sequence of H37Rv isolates, including NR-123, TMC 102 and a culture of the original 1998 isolate (Table 2). Our new H37Rv reference also contains 109 SNPs and 35 indels

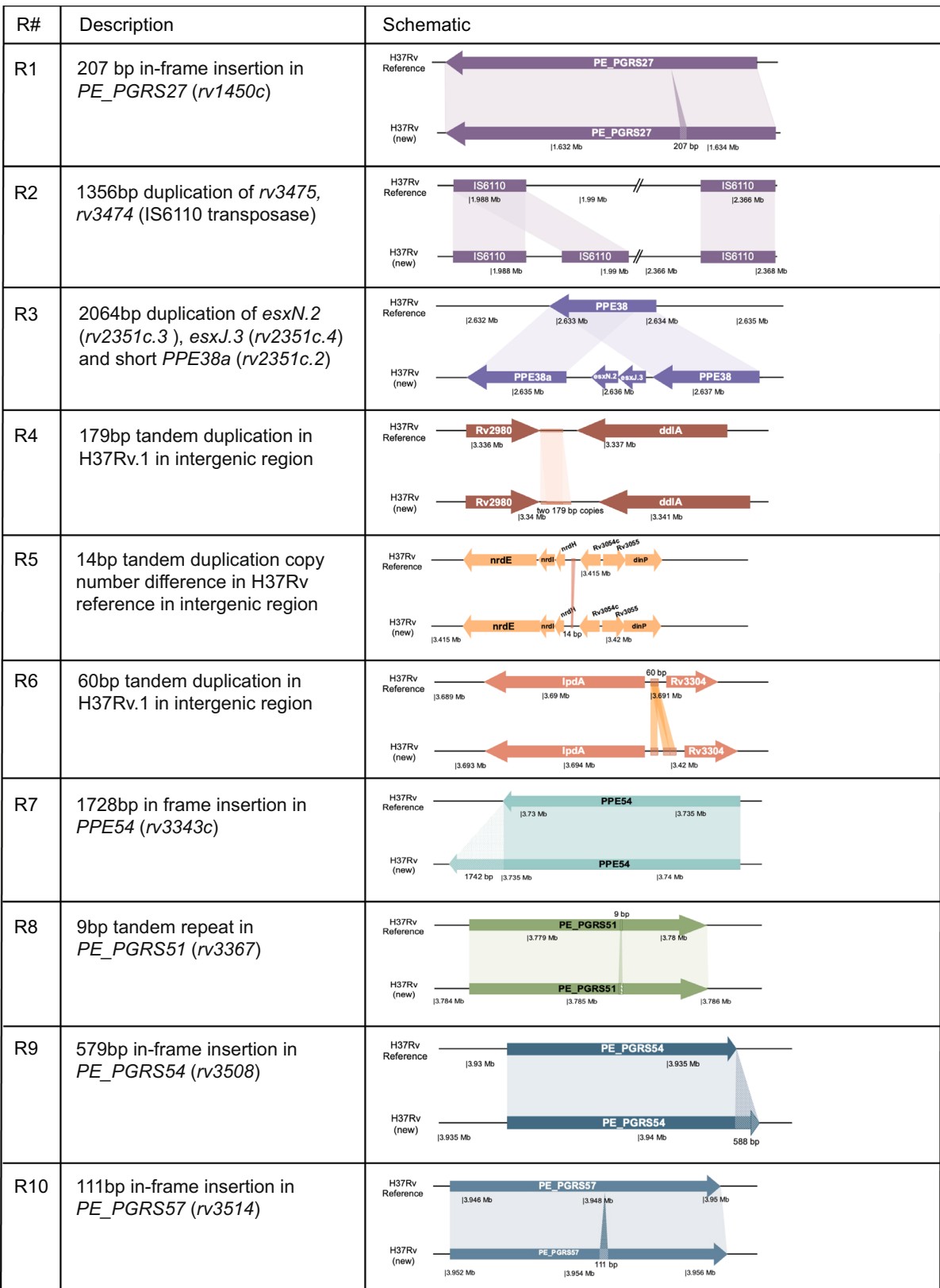

**Fig. 2 | Identifying and validating Regions of difference.** Description of the 10 regions of difference identified by DNAdiff between the H37Rv reference and H37Rv (new).

compared to the Genbank H37Rv reference sequence. We believe that most conclusions reached by previous studies which used the prior reference will remain unchanged since the overall genomic structure of the Genbank H37Rv reference is preserved in our new H37Rv

assemblies. However, our de novo assemblies led to the discovery of new *esxN* and *esxJ* paralogs in H37Rv. These genes are members of the conserved ESX-5 locus. The *esxN* (*rv1793*) gene is a member of the *M. tuberculosis* 9.9 subfamily and is a paralog of *esxA* (ESAT-6). The *esxJ*

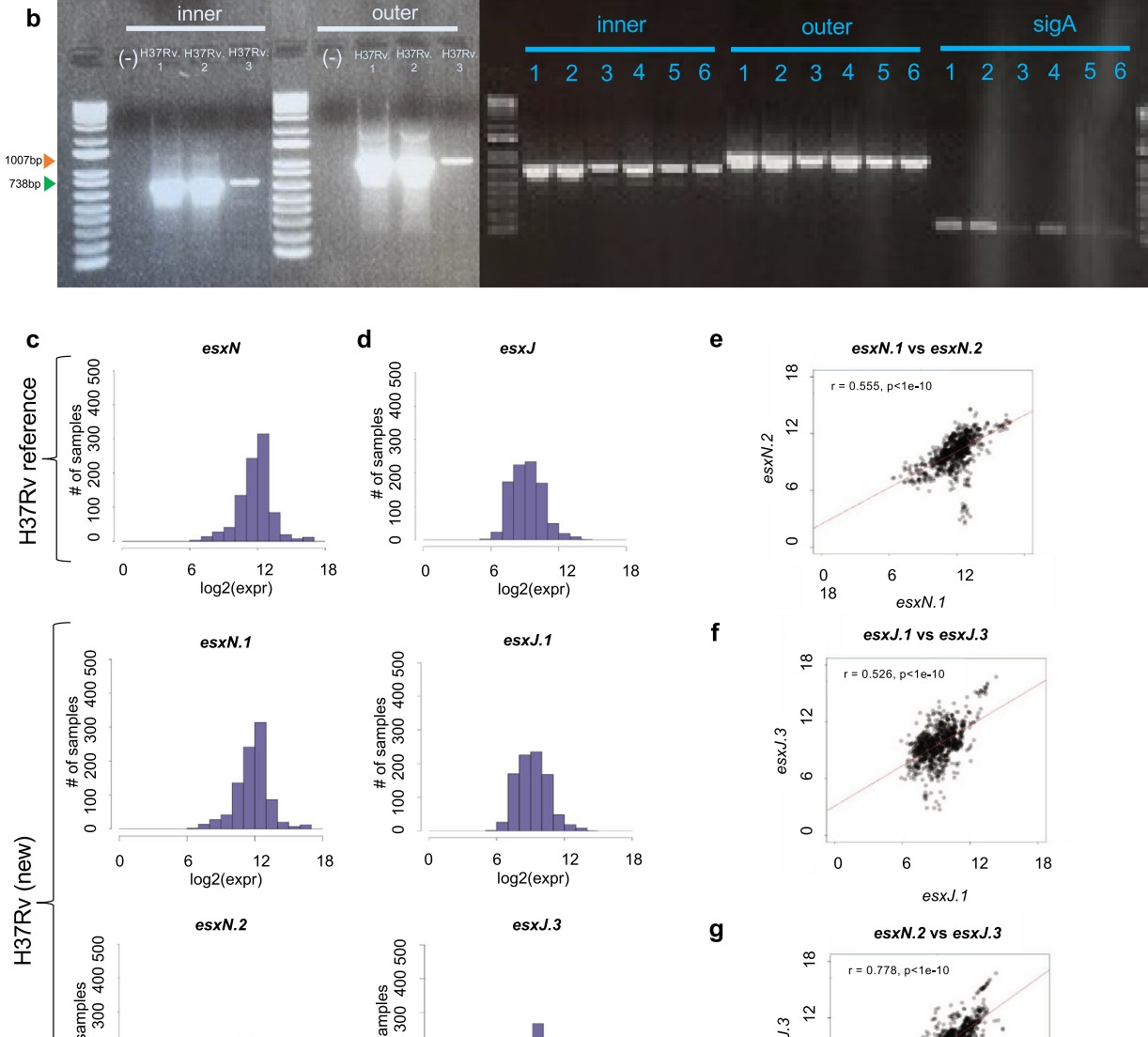

**Fig. 3 | Identifying R3 in all H37Rv strains. a** Clustal Omega comparison of amino acid sequence for *esxN* and *esxN.2* and between *esxJ, esxM* and *esxJ.3*. **b**. PCR of R3 in all 3 laboratory replicates of H37Rv (H37Rv.1-3), H37Rv1998 (1-3) and 3 commercially available strains of H37Rv from ATCC (4: NR-123, 5: TMC102, 6: TMC301). Inner primers targeted inside the region (738 bp) and outer primers targeted flanking regions (1007 bp). *SigA* primers were used as a housekeeping control. **c-d** Histogram of *esxN* (**c**) and *esxJ* (**d**) expression in public H37Rv datasets, demonstrating that newly identified *esxN.2* and *esxJ.3* are expressed in H37Rv and exhibit differential gene expression compared to their respective paralogs. **e-g** Scatterplots showing correlation of expression using Pearsons correlation coefficient in H37Rv (new) of *esxN.1* and *esxN.2* (r = 0.554; p = 7.02e-75) (**e**); *esxJ.1* and *esxJ.3* (r = 0.526; *p* = 1.41e-65) (**f**); and *esxN.2* and *esxJ.3* (r = 0.778; *p* = 2.04e-195) (**g**).

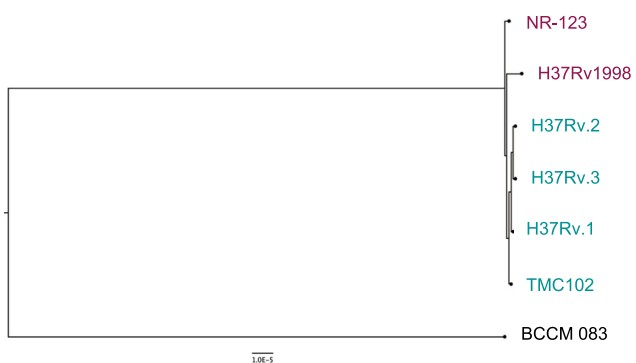

**Fig. 4 | Phylogenetic analysis of H37Rv strains used in the study.** H37Rv strains were assembled with Bact-Builder, sequence alignment was completed using Bowtie2 via REALPHY[68] (https://realphy.unibas.ch/fcgi/realphy), and a maximum likelihood tree was constructed with RaxML[69]. The tree was rooted with BCCM 083, a *M. tuberculosis* Lineage 1 strain[70].

**Table 2 | Description of differences between the newly sequenced and de novo assembled H37Rv strains, TMC102, NR123 and H37Rv1988 identified by DNAdiff**

| R# | Description |
|---|---|
| R11 | 117 bp insertion in *PE_PGRS5* (rv0297) in TMC102 and H37Rv1998 |
| R12 | 123 bp insertion in intergenic region between rv0487 and rv0488 in TMC102 and H37Rv1998 |
| R13 | Additional 1356 bp IS6110 transposon element insertion between *plcD* (rv1755c) and *cut1*(rv1758) in TMC102 |
| R14 | 114 bp insertion in rv2090 (Probable 5′–3′ exonuclease) in TMC102 |
| R15 | 14 bp tandem duplication between *Nrdh* (Rv3053c) and *Rv3054c* in TMC102 |
| R16 | 1356 bp IS6110 transposon element insertion between *higA* (Rv1956) and another IS6110 transposon in TMC102 |
| R17 | 9 bp tandem duplication in *PE_PGRS51* (rv3367) in TMC102 |
| R18 | 51 bp insertion in rv2680 in NR123 and H37Rv1998 |
| R19 | 59 bp insertion in intergenic region between rv2980 and *ddlA* (rv2981c) in H37Rv1998 |
| R20 | 9 bp tandem duplication in *PE_PGRS57* (rv3514) in H37Rv1998 |

(rv1038c) gene is a member of the QILSS subfamily and is a paralog of *esxB* (CFP-10)[31]. Canonical *esxA* and *esxB* proteins interact to form a heterodimer that is secreted from *M. tuberculosis*. These proteins are known to be frequently recognized T-cell antigens[32]. Several *esx* gene pairs are contained within the conserved ESX-1 – ESX-5 loci which together encompass a type VII secretion system (T7SS). ESX-5 has been linked to PE and PPE protein secretion during macrophage infection[33,34]. Loss of ESX-1 has been linked to attenuation in *M. bovis* BCG[35] and there is evidence for attenuation in ESX-5 mutants in *M. tuberculosis* as well[36]. Not only are the new ESX-5 genes which we identified actively transcribed, but they are also differentially expressed compared to the known ESX-5 genes. This strongly suggests that *esxN.2* and *esxJ.3* may have some new functionalities compared to their previously paralogs, including a potential a role in *M. tuberculosis* pathogenesis. Our results also suggest that the RNAseq reads properly attributable to *esxJ.3* may have been misassigned *to esxJ.1* using the current Genbank H37Rv genome reference, potentially confounding analysis of *esx* gene functionality (Figure S4k). Regions with similar genomic organization to R3 were previously observed in *Mycobacterium marinum*[27], *Mycobacterium microti*[29] and *M. tuberculosis* strain CDC1551[27]. However, our findings suggest that the genes previously annotated as *esxX*, *esxY* and *PPE71* should instead be named *esxN.2*, *esxJ.3* and *PPE38* based on sequence homology and PGAP annotation of

our fully closed genome. The *esxJ* gene in the H37Rv reference sequence and our newly annotated *esxJ.3* also share partial sequence homology to *rv1792* which is annotated as *esxM* in the reference genome due to the occurrence of a stop codon in the middle of the gene, substantially shortening the overall protein size. However, *esxM* was annotated as *esxJ* in the *M. tuberculosis* strain CDC1551 annotation despite presence of the stop codon. We have therefore named the new paralog discovered in the H37Rv as *esxJ.3* to avoid confusion, even though there is no *esxJ.2* in the H37Rv1998 reference.

Our new H37Rv reference will also enable more reliable studies of PE/PPE gene function. The *M. tuberculosis* PE/PPE genes are a group of approximately 170 genes that are named for the presence of N-terminal ProGlu and ProProGlu motifs. These genes account for roughly 10% of the *M. tuberculosis* genome[4,37,38]. PE/PPE genes represent a diverse set of genes that have been implicated in *M. tuberculosis* infection, host immune modulation and nutrient transport[28,37,38].

Our updated assembly includes highly reproducible sequences for all the PE/PPE genes. We have also discovered substantial insertions in five known PE/PPE genes and have identified a new PPE gene (*PPE38a*). The functional implications of many PE/PPE genes are unknown, however; further investigation will be made possible with the availability of our newly complete and corrected sequences.

As whole genome sequencing (WGS) becomes increasingly used for studying large communities of related bacteria, accurate reference genomes and methods for de novo whole genome assembly will be needed to avoid sequencing artifacts. Our work demonstrates the importance of using a de novo consensus assembly approach for generating high-quality genomes. These complete and highly accurate assemblies can enhance our understanding of genomic variation and its role in microbial pathogenesis and evolution. Bact-Builder was validated with *M. tuberculosis* which contains a class I genome, like most bacteria[39], and does not contain accessory plasmids. It should be noted that the *M. tuberculosis* genome still exhibits considerable complexity due to the presence of repetitive PE/PPE genes. Bact-Builder users assembling bacteria with accessory plasmids would need to run Trycycler manually, as accessory plasmids/extrachromosomal DNA must be run through Trycycler individually. Future studies will be needed to determine how well Bact-Builder works for performing de novo assemblies of class II and class III bacterial genomes.

Our approach provides several improved and complete reference genomes for the second largest cause of death from an infectious disease. The updated genomes of TMC102, NR123 and H37Rv1998 are quite similar as they all contain R1–R10. However, we did note small differences between each isolate and have deposited updated sequencing data for each (SRA accession #: PRJNA836783). TMC102 appears to be used more commonly used by the global scientific community than NR123, being cited >1000 times versus 246[40], respectively at the time of this writing, suggesting that TMC102 should perhaps become the single accepted laboratory strain for *M. tuberculosis*. Regardless of which H37Rv strain is used in any given experiment, our work suggests that the use of primary ATCC stocks and frequent WGS followed by de novo assembly should be performed to ensure that the genome of the H37Rv being used is sufficiently conserved to allow comparisons between research groups. Bact-Builder may be an invaluable tool for developing comprehensive and accurate genomes, furthering the assembly of a new global set of *M. tuberculosis* reference genomes that together describe a true *M. tuberculosis* pangenome fueling studies that lead to a better understanding of tuberculosis pathogenesis, strain variation, and ultimately new ways to eliminate this age-old disease.

## Methods
### Bacterial strains
Laboratory stocks of *M. tuberculosis* H37Rv (H37Rv.1-3) were used for all real-strain experiments unless otherwise noted. The H37Rv strain

**Table 3 | H37Rv strains used in this study**

| Name | Strain ID | Source | Notes |
|---|---|---|---|
| H37Rv.1-3 | | Laboratory stock, reportedly from the ATCC | |
| NR-123 | ATCC 25618 NR-123 (BEI) | AG Karlson, Trudeau Mycobacterial Culture Collection (TMC) | Derived from E.R. Baldwin's human-lung isolate H37 by W. Steenken |
| TMC 102 | ATCC 27294 | GP Kubica, Dissociated in 1934 | Derived from E.R. Baldwin's human-lung isolate H37 by W. Steenken |
| H37Rv1998 | | Pasteur Institute | H37Rv stock used to generate reference sequence |

(called H37Rv1998 in this work) was the source of the BAC library[41] used to generate the original H37Rv genome sequence described in 1998[4] was obtained from Institute Pasteur, France. Two additional H37Rv strain variants NR-123 (ATCC 25618), and TMC 102 (ATCC 27294), were obtained from BEI Resources[42] and American type culture collection (ATCC)[40] respectively (Table 3).

## Culture and DNA extraction

*M. tuberculosis* strain H37Rv (Table 3) was streaked on 7H11 (Sigma-Aldrich, St. Louis, USA) plates containing: 0.5% (vol/vol) glycerol and 10% oleic acid-albumin-dextrose catalase (OADC) (BD Difco, Frankin Lakes, USA). Single colonies were isolated and cultured in Middlebrook 7H9 (BD Difco, Franklin Lakes, USA) containing: 10% oleic acid-albumin-dextrose catalase (OADC) (BD Difco, Frankin Lakes, USA), 0.05% (vol/vol) Tween 80 (Sigma-Aldrich, St. Louis, USA) and 0.2% (vol/vol) glycerol (Sigma-Aldrich, St. Louis, USA). Late exponential phase cultures were plated on 7H11 (Sigma-Aldrich, St. Louis, USA) plates containing: 0.5% (vol/vol) glycerol and 10% oleic acid-albumin-dextrose catalase (OADC) (BD Difco, Frankin Lakes, USA). Genomic DNA (gDNA) was extracted using a modified Cetyltrimethylammonium bromide (CTAB) method[43]. Briefly, plated strains were grown until confluent (~3 weeks). Roughly 1/3 of the plate (2–3 loopfuls) were added to 400 μl 1x Tris-Ethylenediaminetetraacetic acid (EDTA) pH 7.5 (Thermo Scientific, Waltham, USA) and heated at 80 °C for 30 min. Lysozyme (Sigma-Aldrich, St. Louis, USA) was added to the tube (final concentration 20 mg/ml), followed by incubation at 37 °C overnight. Seventy μl of 10% sodium dodecyl sulfate (SDS) (w/v) (Promega, Madison, USA) and 15 μl proteinase K (20 mg/ml; Invitrogen, Waltham, USA) was added and the sample was incubated at 50 °C for 20 min. A mixture of *N*-acetyl-*N,N,N*-trimethyl ammonium bromide (CTAB; final concentration, 40 mM) and NaCl (final concentration, 0.1 M) was added, followed immediately by the addition of NaCl alone (final concentration, 0.6 M) and the sample was incubated at 50 °C for 10 min, then 700 μl of chloroform-isoamyl alcohol (24:1) (SigmaAldrich, St. Louis, USA) was added. The sample was then pulse vortexed and then centrifuged for 6 min at 16903 xg at room temperature. The upper aqueous phase was transferred to a new tube taking care to avoid removing any part of the interface, then 5 μl of RNAse A (10 mg/ml; Qiagen, Hilden, Germany) was added and the sample was incubated for 30 min at 37 °C. Seven hundred μl of chloroform-isoamyl alcohol (24:1) was then added to the sample for a final extraction. The sample was pulse vortexed and centrifuged for 6 min at 16903 xg at room temperature. The genomic DNA in the resulting aqueous phase was isolated by 1x volume cold isopropanol precipitation. Spooled DNA was removed and washed with 70% cold ethanol. The sample was then spun at 16903 xg for 2 min at 4 °C, the ethanol was removed, and the sample was air dried. The final sample was resuspended in 50 μl nuclease free water.

## Genomic DNA (gDNA) sample quality check

Once extracted, H37Rv gDNA concentration and quality was evaluated using a Nanodrop One™ (ThermoFisher, Waltham, USA). Fragment size and DNA integrity number (DIN score) was evaluated using Genomic DNA Screen Tape on a 4200 TapeStation system

(Agilent, Santa Clara, USA). H37Rv gDNA extracts that met the following cutoffs were used for sequencing: A260/280 ratio: between 1.7–2.0; A260/230 ratio: between 1.5–2.0; DIN score ≥ 7; and DNA length > 20 kb. Purified DNA was stored at 4 °C prior to preparation of sequencing libraries.

## Whole genome sequencing (WGS)

**Nanopore Sequencing.** Three independent replicates of H37Rv laboratory stock (H37Rv.1-3) gDNA were processed with a 1D sequencing kit (SQK-LSK109) (Oxford Nanopore Technologies (ONT, Oxford, United Kingdom)) along with a native barcoding kit (EXP-NBD103 and EXP-NBD112) according to the native barcoding gDNA protocol (Table S1). The gDNA was not sheared but used directly for DNA end repair and ligation. Both ligation steps in the protocol were extended from 10 min to 30 min. To ensure enough library was available for each run, the pooled adapter ligation step was performed in duplicate, and the final libraries were pooled before sequencing. The final library was sequenced through the MinKNOW (v 5.0.5) interface using a FLO-MIN106 flow cell on a MinION instrument.

## Illumina PCR-free sequencing

WGS Illumina reads were used to polish consensus sequences for all samples included in this study. The H37Rv samples were library prepped using an Illumina DNA PCR-free library prep kit (Illumina, San Diego, USA) (Table S1). Sequencing libraries were quality checked using the Qubit™ ssDNA kit (Thermo Fisher, Waltham, USA) and equal volumes were pooled for sequencing. The libraries were then sequenced in a paired-end 150 bp configuration on an Illumina Nova-Seq 6000 platform.

## Artificial in silico read generation

Bact-Builder development and testing made use of artificial reads generated using the H37Rv1998 reference sequence. A published H37Rv fasta obtained from NCBI (NC_000962.3) was run through BadRead (v0.1.5; –quantity 100x -length 1000, 1000) (https://github.com/rrwick/Badread) in order to generate simulated ONT reads[23]. In silico paired end Illumina reads were generated with ART[24] using the same published H37Rv fasta and the default ART parameters to generate Illumina 150 bp paired-end reads with a fold coverage of 30x.

## Assembly pipeline

A full reproducible pipeline encompassing all these steps herein was built within Nextflow[44] and can be found at github.com/alemenze/bact-builder. Parameters and software versions listed below are defaults within the automated pipeline. An overview of the entire pipeline can be found in Fig. 1a.

## Compute resources

The pipeline was run within the local Amarel HPC environment but has been designed to be optimized to custom HPC or cloud-based resources. The local environment consisted of base nodes with 2x Intel Xeon Gold 6230 R (Cascade Lake) Processors (35.75 MB cache, 2.10 GHz): 2933 MHz DDR4 memory, 26-core processors (52 cores/

node), 12×16 GB DIMMS (192 GB/node) per node, and GPU nodes included graphics cards with the NVIDIA Pascal architecture.

## Basecalling and demultiplexing

Raw nanopore sequencing reads were base-called and demultiplexed with the latest version of the Guppy basecaller available at the point of sequencing (Guppy v.4.2.2). Sequencing quality and output was evaluated using Nanoplot (v1.33.0) (https://github.com/wdecoster/NanoPlot)[45] and pycoQC (v2.5.0.23) (https://github.com/tleonardi/pycoQC)[46].

## Quality control and filtering

Basecalled and demultiplexed files were trimmed using Filtlong (v0.2.0; -min_length 1000 -min_mean_q 70) (https://github.com/rrwick/Filtlong). Sample contamination was evaluated via taxonomic classification using Kraken2 (v2.1.1) (https://github.com/DerrickWood/kraken2)[47,48]. Illumina data was trimmed using trim-galore (v0.6.6) (https://github.com/FelixKrueger/TrimGalore).

## Random sub-setting

Nanopore reads were randomly sub-selected to even read depths and variant samples using Rasusa (v0.3) (https://github.com/mbhall88/rasusa).

## Assembly

Nanopore reads for each sample were de novo assembled using the following four assemblies in triplicate: Canu (v1.5; genomeSize= 5 m)[11], Flye (v2.8.1-b1676; genome-size 5 m -plasmids)[12], Miniasm (v0.3-r179)[49], and Raven (v1.3.0)[13]. Gap-closed sequences were evaluated in Bandage (v0.8.1) (https://github.com/rrwick/Bandage)[50].

## Consensus assembly

Trycycler (v0.3.0) (https://github.com/rrwick/Trycycler) was used to generate a consensus long-read assembly using the twelve individual assemblies. The cluster step was run with the following modifications: trycycler cluster -min_contig_depth 0.5. Within Bact-Builder, cluster001 is used by default for downstream processes. Trycycler can be run manually for bacteria with extra-chromosomal plasmids or if assemblers need to be removed. Trycycler reconcile was run with the following modifications: trycycler reconcile -cluster_dir trycycler/cluster_001 --max_length_diff 1.3 -max_add_seq 10000 -min_identity 95 -max_indel_size 1000. All other steps were run with default parameters. Following the reconcile step, contigs that could not be reconciled or had a high degree of dissimilarity from the majority of assemblies were removed and the reconcile step was run again. As long as ≥ 6 assemblies passed the reconcile step, Trycycler was run to completion and the consensus was used for downstream analysis.

## Polishing

Following Trycycler, the consensus assemblies were further polished using both long and short read polishers. Racon (v1.4.20; -m 8 -x −6 -g −8 -w 500) (https://github.com/isovic/racon) was run three times on each assembly using ONT reads. Following racon, medaka (v1.0.1; -m r941_min_high_g360) (https://github.com/nanoporetech/medaka) was used for long read polishing. Finally, Pilon (v1.24; -fix all -changes) (https://github.com/broadinstitute/pilon) was run three times on the medaka consensus using Illumina reads[51].

## Evaluating outputs

Final assemblies were assessed for SNPs, indels and regions of difference using DNAdiff (v1.3) from the MUMmer program (https://github.com/mummer4/mummer)[52,53] and quast (v5.0.2). Complete, polished assemblies were also compared to each other using a pan-genomics approach implemented in anvi'o[54] (https://anvio.org) to investigate large structural changes and differences in the accessory gene pool as a function of the assembly and curation steps. Annotation of the consensus genome sequence was performed using the NCBI's prokaryotic genome annotation pipeline (PGAP)[55].

## PCR and sanger sequencing

Following DNAdiff analysis, identified regions were PCR validated by designing PCR primers both within the identified region and flanking regions. PCR primers were designed using Primer Quest (IDT, Coralville, USA) (Supplementary Data 4). GoTaq Green Master Mix (Promega) was used to run PCR for 30 cycles using manufacturers suggested parameters[56]. PCR products were run on a 1% Agarose gel containing .08% ethidium bromide and visualized using the ChemiDoc imaging system (BioRad, Hercules, USA). PCR amplicons were sanger sequenced (Psomagen, Rockville, USA) and aligned to the reference to validate the region.

## Expression analysis

Accession numbers for all *M. tuberculosis* H37Rv RNA-sequencing experiments publicly available on the NCBI Sequence Read Archive (SRA) as of October 1, 2021 were downloaded using the NCBI's esearch and efetch utilities, similar to Yoo et al.[57] Sequencing runs were filtered to include only those performed with Illumina short read sequencing, and runs were processed as described in Ma et al. 2021[58]. Sequences were downloaded using their SRA accession numbers using fasterq-dump[59]. Raw sequence quality was assessed using fastqc[60] and adapter trimming was performed using bbduk[61]. Trimming and sequence removal based on quality was performed using trimmomatic[62]. Sequences were aligned to either the existing H37Rv reference (NC_000962.3) or to H37Rv(new) using Bowtie2[63]. Read counts were compiled using featureCounts[64]. Quality data, adapter and quality trimming statistics, and alignment and counts metrics were compiled and assessed using multiqc[65]. Of 908 RNA-sequencing runs available from the strain H37Rv, 905 used Illumina sequencing chemistry and passed quality control metrics and were included in the expression analyses reported here. Batch correction was completed by grouping samples by study, identified by the BioSample ID corresponding to a given run's SRA accession, and performing quantile normalization on raw counts of reads using qsmooth[66]. Visualization of data was performed in R[67].

To evaluate whether expression reads mapping to the novel paralogs were incorrectly mapping to *esxN.1* and *esxJ.1*, Bowtie2 aligner with the "-very-sensitive-local" option was used to align triplicate H37Rv RNAseq data (SRA accession: PRJNA507615) against both the *esxN.2* and the *esxJ.3* gene sequence. We then extracted the reads that successfully mapped to each gene and aligned these to the existing H37Rv1998 reference genome. We collected information on the number of reads in each experiment, the number of reads that mapped to each respective gene, and the subset of reads that additionally mapped to the H37Rv genome.

## Reporting summary

Further information on research design is available in the Nature Portfolio Reporting Summary linked to this article.

# Data availability

Data Collection: Whole Genome Sequencing data for the strains used in this study have been deposited in the NCBI Sequence Read Archive via BioProject accession number PRJNA836783. Updated sequence files have been deposited in GenBank, with primary accession code: CP110619 . The Fasta sequence and annotated file for H37Rv(new) can also be found in the Supplementary files as Supplementary Data 2 and Supplementary Data 3 respectively.

## Code availability

Bact-Builder is available at: https://github.com/alemenze/bact-builder. Scripts for the RNASeq analysis can be found at: https://github.com/as2654/rna-seq-tb0.

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

## Acknowledgements

Research reported in this publication was supported by the National Institute of Allergy and Infectious Diseases of the National Institutes of Health under award numbers U19AI11276 and U19AI162598. The content is solely the responsibility of the authors and does not necessarily represent the official views of the National Institutes of Health. Additionally, JHY is supported by the following National Institutes of Health awards: R00-GM118907 and R01-AI146198 and an Agilent Early Career Professor Award. The authors acknowledge the Genomics Center Rutgers New Jersey Medical School (https://research.njms.rutgers.edu/genomics/) and the Office of Advanced Research Computing (OARC) at Rutgers, The State University of New Jersey (http://oarc.rutgers.edu) for providing access to the Amarel cluster and associated research computing resources that have contributed to the results reported here. We thank Dr. Stewart Cole for his advice and support and for his kind gift of the H37Rv1998 strain.

## Author contributions

P.C., P.K. and D.A. designed the study; P.C. and A.D.L. developed the bioinformatics tools; P.C. and C.A.G. library prepped and sequenced the samples; P.C., A.D.L., E.C.F., A.S. and A.O.M. performed data analysis; P.C., E.C.F., A.M.E., and A.O.M. contributed to data visualization. P.C. and D.A. wrote the manuscript; A.D.L., E.C.F., A.S., A.O.M., W.E.J., J.H.Y., A.M.E., R.B. and P.K. edited the manuscript.

## Competing interests

The authors declare no competing interests.
