## [Peer Review File · Nature Communications]

A comprehensive update to the *Mycobacterium tuberculosis* H37Rv reference genomeReviewer #1 (Remarks to the Author):

In this work, Chitale and collaborators re-sequenced and reassembled H37Rv, a laboratory *M. tuberculosis* strain that is the most commonly used reference for genomic studies. The original sequence was derived in 1998 by Stewart Cole et al., and since then it has been used in hundreds of studies. The authors found a relatively low number of differences (10 regions of difference, 109 SNPs and 36 indels) between the new assembly and the original one, pointing to the good quality of the original sequence, even with the technical limitations present in 1998 in comparison with the technology available nowadays. The authors also performed an expression analysis to check if the new regions are expressed or not.

The idea of reviewing and reconstructing the H37Rv genome sequence using current sequencing technologies is good and I think that the work was necessary. The manuscript is well written, results are clearly exposed and methods are detailed in the main text, supp text and the github repository. The software repository is well documented and clearly explained.

I think that this work deserves publication, and I just have a few brief comments that I think could improve some of the parts and that will not require a lot of work.

-The new genomic sequence of the H37Rv reassembly is lacking, as well as the new annotation file. Please, include both in the paper and in a public repository.

-Please, besides description of new RD, include a complete catalogue/table of all the differences found between the new proposed reference and the 1998 one (the 109 SNPs and the 36 indels).

-Regarding the expression analysis, are the sequencing reads that currently map the new regions (esxN.2 and esxJ.3) mapping their 'original' paralogs (esxN and esxJ) in the H37Rv-1998 genome? If the genes are highly duplicated, sequencing reads can be misassigned, hence the expression levels of these genes can be wrongly interpreted.

-Authors should discuss the implications of generalizing the use of the new H37Rv sequence instead of the H37Rv-1998 sequence. This reference has been extensively used for mapping WGS TB data and perform epidemiological analysis, drug-resistance predictions, evolution studies, etc. What would be the implications of changing the reference? Would we be able to compare results and studies performed with different references?

-A color scale indicating the meaning of the green-yellow scale is missing in figure 1B.

Reviewer #2 (Remarks to the Author):

The authors present an assembly pipeline for bacterial genomes and use it to update the reference strain of *Mycobacterium tuberculosis*. They find likely errors in the existing reference and validate them by comparing multiple strains. I think it is a very nice result and keeping/updating references when technology improves is a worthy goal. The description of the changes from the reference and their potential significance was well-described and presented.

My primary concern is with the assembly pipeline. One of the major issues is the components used are extremely out of date (2-5 years old). I'm not clear there are any plans to update them. The repository doesn't appear active, with almost no changes since last year except documentation updates. There have been several similar pipelines developed but none gain traction in the community due to the difficulty of getting started with them and the lack of maintenance or generalizability. In contrast, the genome assemblers used here are frequently updated and tested across a variety of

organisms.

- **The authors state that they can produce a closed genome for any bacterial strain but this likely depends on both the complexity of the genome in question but also the quality of the data. There are no tests on anything besides the relatively simple class I (Koren et al. Genome Biology 2013) Mycobacterium genome.**
- **The authors mention other assembly pipelines but do not compare to them. I would be interested how they perform and if they address some of the instability observed between assemblers.**
- **The authors mention instability between assemblers but both Canu and Flye appear reliable and the differences likely come from circularization. Are the assemblies consistent in length if redundant sequence is removed?**
- **Again, missing genes and other issues are mentioned by it would be interesting to see the breakdown if a canu or flye assembly was polished with racon/medaka/pilon.**
- **Most of the software used is quite old (Flye version is almost 2 yrs old, Canu is over 5 years, medaka is 2 years old, base caller is 1.5 yrs old) so it is also not clear if some of the issues the authors observed have been addressed by the assembler developers.**
- **The pipeline itself is quite heavyweight and I couldn't run it locally due to a lack of suitable input data. I'd suggest the tutorial should at least provide a toy test dataset to verify installation.**

Reviewer #1

In this work, Chitale and collaborators re-sequenced and reassembled H37Rv, a laboratory *M. tuberculosis* strain that is the most commonly used reference for genomic studies. The original sequence was derived in 1998 by Stewart Cole et al., and since then it has been used in hundreds of studies. The authors found a relatively low number of differences (10 regions of difference, 109 SNPs and 36 indels) between the new assembly and the original one, pointing to the good quality of the original sequence, even with the technical limitations present in 1998 in comparison with the technology available nowadays. The authors also performed an expression analysis to check if the new regions are expressed or not. The idea of reviewing and reconstructing the H37Rv genome sequence using current sequencing technologies is good and I think that the work was necessary. The manuscript is well written, results are clearly exposed, and methods are detailed in the main text, supp text and the github repository. The software repository is well documented and clearly explained. I think that this work deserves publication, and I just have a few brief comments that I think could improve some of the parts and that will not require a lot of work.

1. Reviewer: The new genomic sequence of the H37Rv reassembly is lacking, as well as the new annotation file. Please, include both in the paper and in a public repository.

Response: We have now included the H37Rv (new) sequence as a supplementary data file, and the whole genome sequencing data for the strains used in this study are available from the NCBI Sequence Read Archive via BioProject accession number PRJNA475130. Updated annotation files have been deposited in GenBank, with primary accession code: SUB11453622 (GenBank approval pending). The location of these files are now listed in the "Data availability" section of Methods.

2. Reviewer: Please, besides description of new RD, include a complete catalogue/table of all the differences found between the new proposed reference and the 1998 one (the 109 SNPs and the 36 indels).

Response: We have included a complete table of SNP and indels identified between H37Rv (new) and the 1998 reference sequence in supplementary table 4 (see supplementary files).

3. Reviewer: Regarding the expression analysis, are the sequencing reads that currently map the new regions (*esxN.2* and *esxJ.3*) mapping their 'original' paralogs (*esxN* and *esxJ*) in the H37Rv-1998 genome? If the genes are highly duplicated, sequencing reads can be misassigned, hence the expression levels of these genes can be wrongly interpreted.

Response: We thank the reviewer for this interesting question. To address this, we used the Bowtie2 aligner with the "--very-sensitive-local" option to align a complete, sequenced tuberculosis RNAseq experiment done in triplicate (SRA accession # PRJNA507615) against both the *esxN.2* and the *esxJ.3* gene sequence. We then extracted the reads that successfully mapped to each gene and aligned these to the existing H37Rv reference genome. We collected information on the number of reads in each experiment, the number of reads that mapped and did not map to the gene, and the subset of reads that additionally mapped to the H37Rv genome. Our results show that, on average, 0.2% of reads that aligned to the *esxN.2* gene also mapped to the H37Rv genome without being discarded. Meanwhile, on average, 54.5% of reads that aligned to the *esxJ.3* gene also mapped to the H37Rv genome without being discarded (Figure S4 K). This is likely due to high degree of homology between *esxJ.1*, *esxM* and *esxJ.3*. These results strongly suggest that RNAseq reads from *esxN.2* are simply discarded from an analysis based on the standard 1998 H37Rv reference, while a significant number of reads from *esxJ.3*

are improperly attributed to *esxJ.1* using the 1998 H37Rv reference. Neither of these outcomes are desirable and both can be corrected by mapping RNAseq results to our updated H37Rv genome (and eliminating *esx* reads that map to more than one gene). We have included this data in the updated Results and addressed its implications in the Discussion.

4. Reviewer: Authors should discuss the implications of generalizing the use of the new H37Rv sequence instead of the H37Rv-1998 sequence. This reference has been extensively used for mapping WGS TB data and perform epidemiological analysis, drug-resistance predictions, evolution studies, etc. What would be the implications of changing the reference? Would we be able to compare results and studies performed with different references?

Response: The reviewer brings up an important point since the original H37Rv reference has been used for years to align *M. tuberculosis* sequencing data. Our revised sequence demonstrates several differences; however, outside of the 10 identified regions of difference, the structure of the genome largely remains the same and we do not believe that most WGS studies will change as a result of our updates. We now make this point in the Discussion. However, our change in the H37Rv reference sequence can have at least two beneficial impacts which are likely to be substantive. 1) The highly repetitive *M. tuberculosis* PE/PPE genes have been difficult to sequence and assemble and have been historically ignored in most GWAS analysis pipelines. These genes constitute approximately 10% of the entire genome, and thus their omission from most genetic studies may have had a substantial impact in our understanding of functional genetic variation in *M. tuberculosis*. Our highly accurate revised sequence will now allow future studies to include these highly repetitive regions as well as the novel paralogs we identified. We have now included a mention of this improvement in the Discussion. The second impact in of our new reference will be in the additional *esx* genes described in R3 and their effect on RNAseq analysis. In response to reviewer #1's comment #3 above, we have already added a short discussion of the implications of our new H37Rv reference for analyzing *esx* RNAseq data in the Discussion.

5. Reviewer. A color scale indicating the meaning of the green-yellow scale is missing in figure 1B.

Response: The scale should be present in the figure submitted with this manuscript. For the purpose of pointing this out to the reviewer we reproduced the figure below with an added yellow box around the scale. See below.

Reviewer #2

1. Reviewer: The authors present an assembly pipeline for bacterial genomes and use it to update the reference strain of *Mycobacterium tuberculosis*. They find likely errors in the existing reference and validate them by comparing multiple strains. I think it is a very nice result and keeping/updating references when technology improves is a worthy goal. The description of the changes from the reference and their potential significance was well-described and presented. My primary concern is with the assembly pipeline. One of the major issues is the components used are extremely out of date (2-5 years old). I'm not clear there are any plans to update them. The repository doesn't appear active, with almost no changes since last year except documentation updates. There have been several similar pipelines developed but none gain traction in the community due to the difficulty of getting started with them and the lack of maintenance or generalizability. In contrast, the genome assemblers used here are frequently updated and tested across a variety of organisms.

Response: Please see #5 and #6 (below) which specifically address the issues brought up by the above general statement/question.

2. Reviewer. The authors state that they can produce a closed genome for any bacterial strain but this likely depends on both the complexity of the genome in question but also the quality of the data. There are no tests on anything besides the relatively simple class I (Koren et al. *Genome Biology* 2013) *Mycobacterium* genome.

Response: Bact-Builder was tested on H37Rv, the reference genome for *M. tuberculosis*. It should be noted that we have only validated Bact-Builder in this class I genome and Bact-Builder does not work well for bacteria that contain accessory plasmids. The program can still be used for genomes that include accessory plasmids, but the user would have to run the tricycler portion of Bact-Builder manually since accessory plasmids/ extra chromosomal DNA would have to be run through tricycler individually. Regarding the utility of Bact-Builder, we point out that the majority of bacterial genomes (>60%) are class I. We have now added a brief discussion of the limitations mentioned here in our Discussion. We thank the reviewer for giving us the opportunity to clarify these points.

We would also like to note that although *M. tuberculosis* falls under class I genomes by Koren et al.'s classification, *M. tuberculosis* does have a relatively complicated genome. PE/PPE genes for example are highly repetitive genes and represent ~10% of the *M. tuberculosis* genome. Most analysis pipelines disregard them all together because they cannot be reliably assembled. However our ability to generate a single, gap closed genome that includes reproducible PE/PPE gene sequences, investigators can now study these repetitive genes and regions in the same way all other *M. tuberculosis* genes are analyzed.

In response to the reviewer's comments, we now point out in the Discussion that Bact-Builder is designed to work with class I genomes, which represent the majority of bacterial genomes, and its utility on more complex class II and class III genomes remains to be established.

3 Reviewer. The authors mention other assembly pipelines but do not compare to them. I would be interested how they perform and if they address some of the instability observed between assemblers.

Response: The majority of other pipelines use a single assembler followed by some type of short read polishing. We compared each of the individual assemblers + polishing used in Bact-Builder.

As shown in response #5 (below) each assembler followed by polishing produced genomes which contained inconsistencies both with other assemblers and against repeat assemblies performed by the same assembler. We also evaluated Unicycler, a program that builds a scaffold of illumina reads and then completes genomes with ONT long reads and found that Unicycler could not generate a single closed out genome (see table below). Unicycler generated a 2 contig genome that was smaller than our Bact-Builder genome. The Unicycler genome also identified additional SNPs and indels that are likely assembly artifacts. Furthermore, the Unicycler assembly was able to identify the 10 regions of difference correctly but due to the fragmented genome, also identified other regions that do not exist in the genome.

Genome	genome size (bp)	# of contigs	SNPs relative to reference	indels relative to reference	regions of difference
H37Rv ref	4411532	1	n/a	n/a	n/a
H37Rv Bact-Builder	4417941	1	109	36	10
H37Rv Unicycler	4411693	2	188	80	20

We have not included the Unicycler analysis in our revised manuscript since this negative result does not substantially change our conclusions. The results of individual use of the other assemblies tested in #5 below are now included in the Supplement. We would certainly be willing to also include the Unicycler results if the reviewer or editor thinks that this information is important for the reader.

4. Reviewer: The authors mention instability between assemblers but both Canu and Flye appear reliable and the differences likely come from circularization. Are the assemblies consistent in length if redundant sequence is removed?

Response: The assemblers used individually and polished are not substantially larger than the genome size of the Bact-Builder constructed H37Rv(new) (see Table below); therefore the inaccuracies of the individual assemblers are unlikely to be due to redundant sequences.

5. Reviewer: Again, missing genes and other issues are mentioned by it would be interesting to see the breakdown if a canu or flye assembly was polished with racon/medaka/pilon.

Assembly	size	SNP count compared to H37Rv reference	SNP count compared to H37Rv (new)	indel compared to H37Rv reference	indel count compared to H37Rv (new)	Regions of difference relative to H37Rv (new)	Region of difference relative to H37Rv reference
H37Rv ref	4411532	n/a	109	n/a	36	10	0
H37Rv (new)	4417941	109	0	36	0	0	10
Canu 1	4440440	109	0	35	0	0	10
Canu 2	4444608	109	0	38	4	0	10
Canu 3	4443581	109	0	36	0	0	10
Flye 1	4417941	109	0	36	0	0	10
Flye 2	4417942	109	0	35	1	0	10
Flye 3	4417934	109	0	35	1	0	10
Miniasm 1	4417942	109	0	35	1	0	10
Miniasm 2	4417941	109	0	36	0	0	10
Miniasm 3	4417942	109	0	37	1	0	10
Raven 1	4417934	109	0	35	1	0	10
Raven 2	4417942	109	0	35	1	0	10
Raven 3	4417934	109	0	36	0	0	10

Response: We tested each of the assemblers used in Bact-Builder individually, in triplicate. These assemblies were all then polished using racon/medaka/pilon before analysis. The results are shown in the Table above. Although polishing largely improved genome size and SNP count,

we still observed differences across indel count, and genome size for both the within and the between assembler comparisons. Similarly, polishing of individual assemblies also improved identification of regions of difference, however we also continued to observe differences in the size of the regions identified. Because we still saw variation across genome size, indel count and regions of difference with different assemblers, our results indicate that a consensus genome accounts for and overcomes these inherent differences and allows us to generate the most highly accurate genomes.

6. Reviewer: Most of the software used is quite old (Flye version is almost 2 yrs old, Canu is over 5 years, medaka is 2 years old, base caller is 1.5 yrs old) so it is also not clear if some of the issues the authors observed have been addressed by the assembler developers.

Response: We agree that the current version of Bact-Builder does not utilize the newest versions of the individual programs used. However, we tracked updates over the past 2 years and noticed that they did not significantly improve output. Most of the developer updates largely involved improving time and usability of the program. However, to fully address this reviewer's question, we updated all programs that had undergone significant changes (Guppy (v 6.0.1), Canu (v 2.0), Flye (v 2.9)) and compared the outputs to what we had observed before. We saw that individual assemblers still produced genomes of varying sizes and that polishing was still required to generate an accurate assembly. Importantly, the updated Bact-Builder pipeline produced H37Rv genomes that were identical in size and structure to the genome we originally produced in the submitted manuscript (Figure S5). Therefore, we continue to use the results generated by the earlier Bact-Builder pipeline in our resubmission.

It should also be noted that the current program versions are being maintained for legacy projects. However, as sequencing technology and related programs continue to evolve, we plan to maintain Bact-Builder and create updated versions on a yearly basis.

7. Reviewer: The pipeline itself is quite heavyweight and I couldn't run it locally due to a lack of suitable input data. I'd suggest the tutorial should at least provide a toy test dataset to verify installation.

Response: Bact-Builder contains some very computationally heavy programs, and although it is technically capable of execution in flexible containerized environments, we strongly recommend that it should be run in a high-performance computer system due to the computational demands.

All sequencing data and resulting fasta files are now available from the NCBI Sequence Read Archive via BioProject accession number PRJNA475130. Unfortunately, due to the size of the raw data files, we are unable to provide a test data set on GitHub. However, if the reviewer wishes to test Bact-Builder on their own high performance computer system, they may access the raw data using the SRA Reviewer token:

<https://dataview.ncbi.nlm.nih.gov/object/PRJNA836783?reviewer=6d1hdcd5b3scv2lrao32hvh4e>

Reviewer #1 (Remarks to the Author):

The authors have addressed all my questions so I recommend publication of the manuscript.

Reviewer #2 (Remarks to the Author):

The authors have addressed the majority of my comments. The new table in the response (showing post-polishing assembly stats) should be included in the supplementary materials and mentioned in the discussion. It shows the variation observed between assemblers and replicates was due to lower-quality consensus and not larger assembly differences. The variation between assemblies is almost non-existent, with all having no SNP differences and no novel region differences and usually 0 or 1 indels. The sizes are also consistent, within 10bp of each other (with the exception of Canu which likely needs to be trimmed to remove redundant sequence due to the circular chromosome).

In fact, the consistent sizes (either matching the Bact-Builder assembly or being 1bp greater or 8bp shorter) makes me wonder how certain the authors are that their assembly result is the only right one for the data. Is it possible there is some heterogeneity in the sample leading to the observed differences? Have the authors checked read support for the differences between assemblies to confirm it is an error and not a low-frequency variant? This should be easy to check, for example, is the indel in the 4417942 bp assemblies in a consistent position and a consistent base between all assemblies? What do read mappings look like over that region? This is an important question I think as it may indicate a single answer isn't an accurate representation of the true genomic structure.

Reviewer #2 (Remarks to the Author):

The authors have addressed the majority of my comments. The new table in the response (showing post-polishing assembly stats) should be included in the supplementary materials and mentioned in the discussion. It shows the variation observed between assemblers and replicates was due to lower-quality consensus and not larger assembly differences. The variation between assemblies is almost non-existent, with all having no SNP differences and no novel region differences and usually 0 or 1 indels. The sizes are also consistent, within 10bp of each other (with the exception of Canu which likely needs to be trimmed to remove redundant sequence due to the circular chromosome).

In fact, the consistent sizes (either matching the Bact-Builder assembly or being 1bp greater or 8bp shorter) makes me wonder how certain the authors are that their assembly result is the only right one for the data. Is it possible there is some heterogeneity in the sample leading to the observed differences? Have the authors checked read support for the differences between assemblies to confirm it is an error and not a low-frequency variant? This should be easy to check, for example, is the indel in the 4417942 bp assemblies in a consistent position and a consistent base between all assemblies? What do read mappings look like over that region? This is an important question I think as it may indicate a single answer isn't an accurate representation of the true genomic structure.

We thank Reviewer #2 for their thoughtful review of our work and for their additional questions and comments that have further improved this manuscript.

1. Reviewer: The new table in the response (showing post-polishing assembly stats) should be included in the supplementary materials and mentioned in the discussion. It shows the variation observed between assemblers and replicates was due to lower-quality consensus and not larger assembly differences.

Response: We have now included this table in our supplementary materials (Supplementary table 5).

However, we wish to point out that the table does not “show that the variation between assemblies is simply due to a lower quality consensus”. This is because in addition to the indels shown in this table, we also show that the different assemblies sometimes produced genomes whose size varied substantially. For example, Canu produced assemblies that were on average 25,000bp larger than H37Rv.1. A closer examination of this region of difference showed that it represented an assembly artifact duplication of multiple genes at the beginning and end of the Canu assembly (likely the result of circularization, as the reviewer suggested). Differences of this nature could be highly problematic in larger pangenome studies among different clinical strains, and their avoidance was a principal motivation for our development of Bact-builder.

2. Reviewer: Is it possible there is some heterogeneity in the sample leading to the observed differences?

Response: All of the samples sequenced during this experiment were single colony isolated from streaked plates and then grown in media prior to plating for gDNA extraction. The methods section of the paper has been updated to describe this. Given their single colony origin, we believe that it is highly unlikely that minor variants contribute to heterogeneity in the cells that were extracted. Also, please see response to question #3 below.

3. Reviewer: Have the authors checked read support for the differences between assemblies to confirm it is an error and not a low-frequency variant? This should be easy to check, for example, is the indel in the 4417942 bp assemblies in a consistent position and a consistent base between all assemblies? What do read mappings look like over that region?

Response: We observed that 4 out of the 12 assemblies were 4,417,942 bps in length. Each of these assemblies, as well as 3 additional assemblies (which varied in size) contained a one bp C insertion at position 3,251,952 compared to our H37Rv.1 sequence (see table below, also supplementary table 6). Oxford Nanopore Technologies (ONT) and illumina data were mapped to H37Rv.1, Canu 1 and Canu 2 polished assemblies using minimap2 and bowtie2 respectively. Canu 1 and Canu 2 were chosen because the indels found in those assemblies represented all of the indels found across all 12 assemblies (See table below also Supplementary table 6). Evaluation of ONT read coverage at Canu 1 1,118,735insC and Canu 2 3,583,601insC, which were the same basepair deleted in H37Rv.1, indicated that the insertion was present (see Figure below, also supplementary figure 5) in the reads and likely not the result of a minor variant present in the sample. Furthermore, evaluation of the illumina paired end reads indicated that there was sufficient coverage at the indel site (see figure below, also supplementary figure 5) indicating that the base was truly present in the majority of the sample and again not the result of a minor variant. This Figure and indel positions are now included in the supplementary section of the paper.

This “false” deletion in our H37Rv.1 genome was the result of our choosing H37Rv.1 as the reference sequence for H37Rv(new). As we state in our manuscript, we also used Bact-Builder to assemble 2 additional replicates (H37Rv.2 and H37Rv.3). Both of these additional replicates were 1 one bp longer H37Rv.1. Closer examination of these sequences revealed that the 3,251,952insC was present in both H37Rv.2 and H37Rv.3, suggesting that this deletion in H37Rv.1 was simply an assembly error in the H37Rv.1 sequence. Following the reviewer’s advice, we updated Bact-builder v1.0 to Bact-builder v1.1 by updating the Guppy basecaller (v4.2.2 -> 6.0.1), Canu (v1.5 ->2.0) and Flye (v2.8 -> 2.9) assemblers. Reassembling H37Rv.1 with Bact-builder v1.1 now includes the 3,251,952insC found in H37Rv.2 and H37Rv.3. We thank the reviewer for suggesting this update to Bact builder. We will upload the updated H37Rv(new) sequence to GenBank, and have included the fasta in the supplementary files of the paper.

To be thorough, we also evaluated the other identified indels found in the Canu 2 assembly relative to H37Rv.1 (see below also supplementary table 6). Two tandem base insertions: 2,313,498insG and 2,313,499insA were present were present in multiple reads in the Canu 2 ONT assemblies; however, only 7% of the illumina reads aligned to this region showed these insertions.

Given the almost complete lack of illumina coverage for these Canu 2 insertions, as well as the absence of these insertions in the Canu 1 and Canu 3 assemblies, the evidence indicates that these two insertions are assembly errors in Canu rather than true minor variants. Furthermore, analysis of ONT and illumina read coverage at H37Rv 104,106insG, revealed that the base was truly present in the sequence and that the deletion in Canu 2 was an assembly error and not a low frequency variant. We now mention these findings in the Results section of the manuscript and refer to a more detailed analysis in the Supplement.

assembly	Position in H37Rv.1	Basepair in H37Rv.1	Basepair in polished assembl	Position in Polished assembly
canu 1	3251952	.	C	1118735
canu2	104106	G	.	2313506
	104113	.	G	2313498
	104113	.	A	2313499
	3251952	.	C	3583601

Reviewer #2 (Remarks to the Author):

The authors have addressed my comments, thank you for the detailed response.